# Investigating point-of-care diagnostics for sexually transmitted infections and antimicrobial resistance in antenatal care in Zimbabwe (IPSAZ): protocol for a mixed-methods study

Kevin Martin ![ORCID],[1,2,3] Chido Dziva Chikwari ![ORCID],[2,4] Ethel Dauya,[2] Constance R S Mackworth-Young ![ORCID],[2,5] David Bath,[5] Joseph Tucker,[1] Victoria Simms ![ORCID],[2,4] Tsitsi Bandason,[2] Francis Ndowa,[6] Leolin Katsidzira,[7] Owen Mugurungi,[8] Anna Machiha,[8] Michael Marks ![ORCID],[1,9,10] Katharina Kranzer,[1,2,11] Rashida Ferrand[1,2]

For numbered affiliations see end of article.

**Correspondence to**
Dr Kevin Martin;
kevin.martin@lshtm.ac.uk

## ABSTRACT

**Introduction** Sexually transmitted infections (STIs) can cause serious morbidity, including pelvic inflammatory disease, and adverse pregnancy outcomes. In low/middle-income countries, limited laboratory infrastructure has resulted in a syndrome-based approach being used for management of STIs, which has poor sensitivity and specificity, leading to considerable underdiagnosis and overtreatment. The WHO has called for development and evaluation of strategies to inform replacement of syndromic management by diagnostic testing.
The aim of this project is to evaluate a strategy of point-of-care testing for six STIs in antenatal care (ANC) in Zimbabwe.

**Methods and analysis** A prospective interventional study will be conducted in ANC clinics in Harare province, Zimbabwe. One thousand pregnant women will be recruited when registering for routine ANC. Alongside routine HIV and syphilis testing, participants will be offered an integrated screening package including testing for *Chlamydia trachomatis* (CT), *Neisseria gonorrhoeae* (NG), *Trichomonas vaginalis* (TV) and hepatitis B. All individuals with STIs will receive treatment, partner notification services, risk reduction counselling and referral if needed according to national guidelines. Gonorrhoea samples will be cultured and tested for antimicrobial resistance as per WHO enhanced gonococcal antimicrobial surveillance programme guidelines.
The primary outcome measure is the composite prevalence of CT, NG, TV, syphilis and hepatitis B. A mixed-methods process evaluation and economic evaluation will be conducted to understand the acceptability, feasibility and cost-effectiveness of integrated STI testing, compared with standard of care (syndromic management).

**Ethics and dissemination** The study protocol was approved by the Medical Research Council of Zimbabwe, the Biomedical Research and Training Institute Institutional Review Board, and the London School of Hygiene & Tropical Medicine Research Ethics Committee. Results will be submitted to open-access peer-reviewed

## STRENGTHS AND LIMITATIONS OF THIS STUDY

⇒ The mixed-methods approach, with qualitative and quantitative data, will allow for the development of a multilayered understanding of the acceptability and feasibility of this integrated screening package.
⇒ Inclusion of an economic evaluation will allow for estimation of the cost-effectiveness of this screening package compared with routine care, which is essential for considering the scalability and sustainability of the programme.
⇒ The large sample size will ensure an estimation of sexually transmitted infection (STI) prevalence with high precision among antenatal attendees in Harare.
⇒ This study does not include a formal outcome evaluation to assess the impact of this integrated screening package for STIs on adverse birth outcomes, which will need to be informed by future studies.
⇒ Our focus on urban clinics will likely limit the generalisability of our findings to other urban centres in Southern Africa.

journals, presented at academic meetings and shared with participating communities and with national and international policymaking bodies.
**Trial registration number** NCT05541081

## INTRODUCTION

Globally, there were an estimated 374 million infections of *Chlamydia trachomatis* (CT), *Neisseria gonorrhoeae* (NG), *Trichomonas vaginalis* (TV) and syphilis in 2020 among people aged 15–49 years.[1] Untreated sexually transmitted infections (STIs) can cause adverse pregnancy outcomes, congenital infection and pelvic inflammatory disease.[2] STIs are also associated with increased risk of both HIV transmission and acquisition.[3]

There are multiple contributing factors to the persistence of the high incidence of STIs globally. Complex sociocultural barriers such as stigma, limited sexual health education and barriers to condom use, and biomedical factors such as asymptomatic infections and increasing levels of antimicrobial resistance (AMR) have been exacerbated in recent years by a substandard global response to STIs, characterised by a lack of funding and political commitment.[1]

Effective management of STIs in low/middle-income countries (LMICs) is additionally hindered by the use of syndromic management, which is the provision of treatment to an individual presenting with symptoms and/or signs that may be caused by an STI.[4] This is problematic as the majority of curable STIs are asymptomatic, particularly in women, and are missed by syndromic management.[5] Furthermore, treatment for infections that patients may not have may lead to side effects and increased AMR.[6]

Diagnostic platforms that do not require complicated laboratory infrastructure are available, but cost and lack of evidence on how they should be implemented to maximise both clinical effectiveness and cost-effectiveness in LMICs limit their implementation. Integration of diagnostic STI testing into health systems is likely to be key to reducing rates of STIs in LMICs, and the WHO has called for evidence to inform replacement of syndromic management by diagnostic testing.[7 8]

A group that may particularly benefit from the introduction of diagnostic testing for STIs is pregnant women. Pregnant women in LMICs are a high-risk population for STIs,[9] and diagnosis and treatment may prevent adverse pregnancy outcomes and congenital transmission of some infections.[10] As point-of-care (POC) testing for HIV and syphilis has already been integrated into antenatal care (ANC) services, this provides a platform for further STI testing and management and potentially enhances operational feasibility. In addition to curable STIs, this is also pertinent for hepatitis B, as testing is noted to be essential for the WHO's triple elimination initiative, which aims to eliminate vertical transmission of HIV, syphilis and hepatitis B.[11] Key to triple elimination is the provision of a multidisciplinary approach within routine ANC services.

There are limited data on STI prevalence among pregnant women in Zimbabwe. Prevalence of CT up to 26.0%, of NG up to 6.4% and TV up to 24.8% has been reported in pregnant women in South Africa and Zambia.[12–16] A 2010 study found a TV prevalence of 11.8% and syphilis prevalence of 1.2% among pregnant women in Harare.[17] Recent studies among female youth in Harare have demonstrated a combined CT/NG prevalence between 18.2% and 19.5%.[18 19] A prevalence of hepatitis B ranging between 3.1% and 5.3% has been reported in pregnant women in South Africa.[20–22]

Control of STIs, particularly NG, is additionally compromised by AMR.[23] Surveillance is key to identifying and monitoring AMR. WHO and the Centers for Disease Control, USA established the Enhanced Global Gonococcal Antimicrobial Surveillance Programme (EGASP) in 2015.[24] A sample of at least 100 gonococcal isolates per year per country is recommended.[23] However, gonococcal AMR data are still extremely limited and in 2018, only 5 of 47 countries in the WHO African Region reported susceptibility testing for NG of at least one of ceftriaxone, cefixime, ciprofloxacin and azithromycin.[25–28]

### Rationale

Evidence is required to inform the use of diagnostic testing for STIs in LMICs, at both national and international levels, particularly regarding acceptability, feasibility and cost-effectiveness. Additionally, there is a data gap regarding the prevalence of STIs among pregnant women in Zimbabwe. Given the paucity of data on AMR in NG in Africa, there is also a need to strengthen AMR surveillance systems.

### Aims and objectives

The overall aims of this study are to implement and evaluate a strategy for integration of POC diagnostics for STIs into ANC settings and to establish a gonococcal AMR surveillance strategy aligned with EGASP in Zimbabwe.

The objectives are to:

1. Determine the prevalence and yield of POC testing for CT, NG, TV, syphilis and hepatitis B, and factors associated with presence of STIs among pregnant women.
2. Conduct a mixed-methods process evaluation to understand the acceptability and feasibility of POC STI testing and comprehensive case management in ANC settings.
3. Estimate the cost and cost-effectiveness of integrated STI testing compared with standard of care.
4. Investigate the prevalence of AMR for NG to inform the development of an EGASP in Zimbabwe.

## METHODS AND ANALYSIS

### Study design and setting

A prospective interventional study will be conducted in primary healthcare clinics (PHCs) in Harare province, Zimbabwe. The PHCs are all based in urban, high-density settings, and provide nurse-led services including ANC and uncomplicated deliveries. High-risk women receive their ANC at central hospitals, with referral also available if complications develop in labour. Opt-out HIV and syphilis testing using rapid diagnostic tests is part of routine care. GeneXpert devices are often available for tuberculosis diagnosis, but the study will provide an additional machine to ensure that sufficient diagnostic capacity is available. We previously demonstrated the feasibility of using non-laboratory technicians to operate the GeneXpert device for on-site CT/NG testing in community settings in Bulawayo, Zimbabwe.[29]

### Study population and recruitment

Pregnant women will be recruited when registering for routine ANC, starting in January 2023. It is the intention that only pregnant women attending their first ANC visit of this pregnancy will be recruited. However, if there is

ongoing slower-than-expected recruitment, pregnant women attending for ANC follow-up visits will also be considered for enrolment. There will be no age cut-off for enrolment. Exclusion criteria will be enrolment in this study on a previous antenatal visit and being unable or unwilling to provide written informed consent. Recruitment will be conducted during weekdays only. Pregnant women will be consecutively enrolled as testing capacity allows.

Reasons for declining participation, and for exclusion, will be documented. If participants only consent to some of the STI tests, reasons for declining the others will be recorded.

## Study procedures

The full schedule of events for pregnant women is described in table 1. Following consent, an interviewer-administered questionnaire will collect sociodemographic data, clinical history including STI symptoms and recent antibiotic use, sexual and obstetric history. Contact information will be collected for follow-up.

Participants will provide three self-taken or provider-taken vaginal swab samples. One vaginal swab sample will be tested for CT and NG using the Xpert CT/NG assay (Cepheid), which has an analytical time of 90 min. The GeneXpert device will be operated using a rechargeable powerpack to provide an uninterrupted power supply.

**Table 1** Schedule of events for pregnant women

| Participants | Activity | Day 0 (study entry) | Days 1–5 | Days 4–14 | On partner attendance | Birth | Post partum (planned telephone follow-up) |
|---|---|---|---|---|---|---|---|
| All | Informed consent | X | | | | | |
| All | Questionnaire | X | | | | | |
| All | HIV testing | X | | | | | |
| All | Syphilis testing | X | | | | | |
| All | HBV testing | X | | | | | |
| All | Vaginal swab collection | X | | | | | |
| All | CT/NG testing | X | | | | | |
| All | TV testing | X | | | | | |
| STI test positive | Health education | X | | | | | |
| HBV test positive | Venepuncture | X | | | | | |
| HBV test positive | HBV viral load and ALT testing | X | | | | | |
| HBV test positive | Referral to secondary care | X | | | | | |
| HIV test positive | Referral as per PHC processes | X | | | | | |
| CT/NG/TV/syphilis test positive | Provision of treatment | X | | | | | |
| CT/NG/TV/syphilis test positive | Partner notification advice and slip | X | | | | | |
| NG test positive | Cervical swab collection | X | | | | | |
| NG test positive | Plating of cervical swab and incubation at laboratory | X | | | | | |
| STI test positive Not treated on day 0 | Contact participant by telephone and ask to return to PHC for treatment | | X | | | | |
| Cultured NG isolate | Storage of isolate at −80°C | | | X | | | |
| Partners | Provision of treatment to partners who attend PHC | | | | X | | |
| HBV test positive | Provision of HBV birth dose vaccine coordinated with PHC and secondary care | | | | | X | |
| CT/NG/TV/syphilis test positive | Contact by telephone to collect data on partner notification process | | | | | | X |
| All | Contact by telephone to collect birth outcome data | | | | | | X |
| All | Review of birth registry records to supplement birth outcome data from participants | | | | | | X |

ALT, alanine aminotransferase; CT, *Chlamydia trachomatis*; HBV, hepatitis B virus; NG, *Neisseria gonorrhoeae*; PHC, primary healthcare clinic; STI, sexually transmitted infection; TV, *Trichomonas vaginalis*.

The second swab will be tested for TV using the OSOM Trichomonas Rapid Test (Sekisui Diagnostics), which has an analytical time of 10 min.[30] The third swab will be stored for future studies including possible whole-genome sequencing. A fingerprick blood sample will be taken for HIV, syphilis and hepatitis B testing using the SD BIOLINE HIV/Syphilis Duo (Abbott Diagnostics Medical Co) (analytical time 20 min) and HBsAg 2 (Abbott Diagnostics Medical Co) (analytical time 30 min) rapid tests, respectively.[31] HIV and syphilis testing, referral to HIV services, syphilis treatment and partner notification for those with HIV or syphilis are already part of routine care. The study team will work with health facility staff to integrate the additional STI testing with routine ANC services to prevent duplication of procedures.

Participants with positive test results and their partners will be managed in line with Zimbabwe national treatment guidelines.[32] For participants with an STI syndrome on presentation, immediate treatment will be provided for syndromes such as pelvic inflammatory disease and genital ulcer disease, where testing will not alter management. For vaginal discharge syndrome, participants will ideally wait for their results to receive tailored treatment; however, they will receive metronidazole regardless of results, in order to cover for bacterial vaginosis. For symptomatic participants not willing or able to wait or return for their results, they will have the option to receive full syndromic treatment.

Participants will ideally collect their results and receive treatment if necessary within the same clinical visit. Participants who test positive for an STI but are unable to receive same-day treatment will be actively followed up by telephone, up to five times over a 28-day period, to advise them to return for treatment.

A client-referral approach will be used for notification of sexual partners. Women will be counselled on the importance of their partners receiving treatment and given partner notification (PN) slips for their partners to return for presumptive treatment. Although partners will be able to attend any clinic, treatment will be provided free of charge if they return to the study clinic.

Women newly diagnosed with HIV will be referred for antiretroviral therapy as per local PHC processes. Women newly diagnosed with hepatitis B will have hepatitis B viral load and alanine aminotransferase testing, alongside referral to a gastroenterology specialist in secondary care. Although hepatitis B vaccination is currently included in the Zimbabwe national vaccination schedule at 6, 10 and 14 weeks, birth dose vaccination is not yet standard of care in Zimbabwe. Birth dose vaccines will therefore be provided by the IPSAZ Study. This is likely to be logistically complex and bespoke strategies for implementation will be designed in conjunction with healthcare teams at the individual PHCs and with local secondary care providers.

Birth outcome data will be collected from birth registers, including birth weight, gestation, mode of delivery and stillbirth. Estimated due date, which will be compared with actual birth date to determine prematurity, will be based on last menstrual period. Participants will also be contacted by telephone post partum to facilitate this process by providing information on date and location of birth, and to provide supplemental data if necessary if the birth register is incomplete. Participants will also be asked about number of ANC visits, and for women with positive STI results, if they gave the PN slip to their partner, and if their partners were treated.

## Process evaluation

A mixed-methods process evaluation will be conducted, based on the Medical Research Council (MRC) Process Evaluation Framework.[33] The focus will be on understanding what was implemented and how; how the intervention led to change; and how local context affects implementation and shapes outcomes. Linnan and Steckler's process evaluation framework has also guided the choice of specific research domains related to implementation, where fidelity, dose and reach/coverage are central features.[34]

Table 2 details the process evaluation research domains and questions. A logic model demonstrating the proposed theory of change is shown in figure 1.

Following initial qualitative formative work to refine the testing strategy, a concurrent triangulation strategy will be used, with quantitative and qualitative data collected in parallel, with similar weighting given to each.[35] This will allow for triangulation of data in order to comprehensively address the process evaluation questions. Routine monitoring data will include uptake of testing, treatment and PN, as well as stock monitoring, debriefing minutes and recording of GeneXpert error codes. Data collection will also include structured and unstructured observation, and focus group discussions and in-depth interviews with key stakeholders including pregnant women, partners, clinic staff, policymakers and the research team involved in delivery of the intervention (including both the clinic-based team and laboratory staff). Different topic guides will be designed for interviews and focus group discussions at different stages of implementation, to reflect the changing focus of the process evaluation. Participants for interviews and group discussions will be purposively selected to ensure a relevant range of views for each stage of the process evaluation. Adaptations will be made to the testing strategy based on interim process evaluation findings to improve ongoing implementation.

Thematic analysis on qualitative data will be performed to identify and develop key themes and concepts on addressing what was implemented and how; how the intervention led to change; and how local context affects implementation and shapes outcomes.

## Economic evaluation

Cost-effectiveness will be evaluated as the ratio of incremental costs and incremental effects of POC testing for CT, NG and TV, in comparison with routine care using syndromic management. Total costs and effects will be

**Table 2** Framework for process evaluation

| Framework domain | Research domain | Research questions | Data collection methods and sources |
|---|---|---|---|
| Implementation | Fidelity | ► How did implementation vary from the protocol that is (a) offering STI screening, (b) undertaking STI screening, (c) providing comprehensive case management including partner notification, (d) training and supervision of staff?<br>► What were the barriers and facilitators to implementation?<br>► What adaptations were made? | Routine monitoring data<br>Structured observation<br>FGDs with clinic staff (1–2 per PHC) and IDIs with research team (all members), pregnant women (8–10 per PHC), partners (3–4 per PHC) and clinic staff (4 per PHC)*<br>IDIs with pregnant women who decline STI screening (3–4 per PHC) to explore reasons for this |
| | Coverage | ► What proportion of: (a) pregnant women attending antenatal care were offered STI screening, (b) pregnant women who were offered STI screening took it up, (c) positive STI cases were treated, (d) partners of positive cases were treated?<br>► How equitable was this coverage?<br>► What were the barriers and facilitators to each step? | |
| Mechanisms of impact | Responses to and interactions with the intervention | ► Which components of the intervention were best accepted and adopted by pregnant women and HCWs and why?<br>► What challenges and barriers were faced? | Structured observation<br>FGDs and IDIs with pregnant women, partners and HCWs* |
| | Interactions and consequences | ► How did various components of the intervention interact?<br>► Were there any unanticipated pathways or consequences? | |
| Context | Proximal and distal | ► What social, cultural, political and logistical factors impede or facilitate how the intervention was implemented, and how were HCWs able to engage with and adopt aspects of the intervention?<br>► What were contextual reasons for adaptations to the intervention and its delivery? | FGDs and IDIs with pregnant women, partners and HCWs*<br>Structured and unstructured observations<br>Key informant interviews with local health authorities and community leaders<br>Context diaries to record external events |

*Number of IDIs/FGDs listed is approximation.
FGDs, focus group discussions; HCWs, healthcare workers; IDIs, in-depth interviews; PHC, primary healthcare clinic; STI, sexually transmitted infection.

estimated for each arm, with incremental values calculated from these estimates.

Clinical endpoints include mild infection, pelvic inflammatory disease, infertility and adverse birth outcomes including miscarriage, stillbirth, low birth weight and prematurity. Incidence of clinical endpoints will be estimated from existing literature, with such estimates for adverse birth outcomes being supported by primary data collection. Disability-adjusted life years will be the main outcome, and will be modelled based on clinical endpoints, predicted life expectancy and disability weights.[36] Intermediate measures include cost per patient screened, diagnosed with an STI and treated. A simple static decision tree model will be used to structure the cost-effectiveness evaluation, which will be from a health system perspective.

Health system costs required to deliver the testing strategy will be collected from both primary and secondary data sources using a bespoke cost extraction tool. Financial and economic costs will be estimated based on study financial records, staff interviews, time-and-motion studies and other sources such as national salary scales for the Zimbabwe health service. Cost categories will include start-up costs such as training costs, capital costs including buildings and storage and equipment, and recurrent costs, including personnel, consumables, and transport

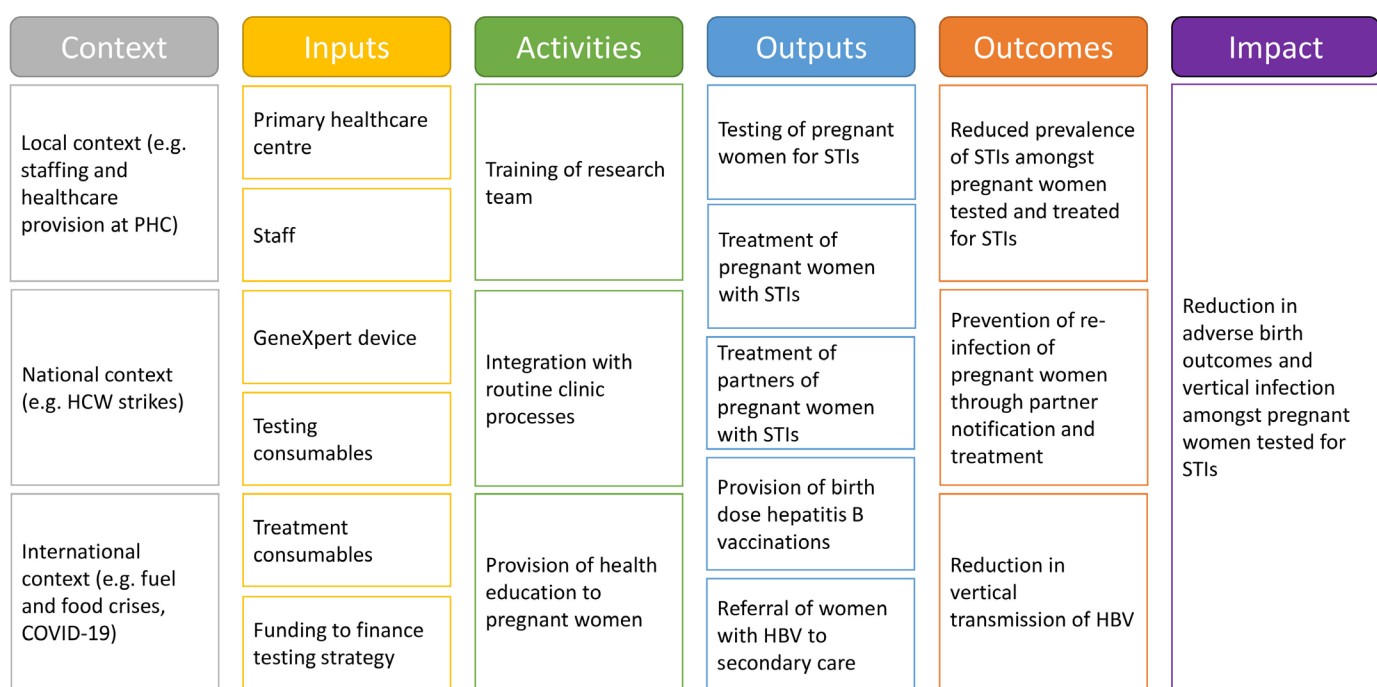

**Figure 1** Logic model for mixed-methods process evaluation. HBV, hepatitis B virus; HCW, healthcare worker; PHC, primary healthcare clinic; STIs, sexually transmitted infections.

costs. Costs associated with downstream complications of infection will be estimated from the literature. Costs incurred by patients while attending ANC, including transport costs and opportunity costs, will be collected using the interviewer-administered questionnaire. Additional costs associated with the testing strategy, such as a longer visit or to return for treatment, will be recorded.

Individual and combined parameter uncertainty will be investigated using deterministic and probabilistic sensitivity analyses. The cost-effectiveness of the testing strategy compared with syndromic management will be compared against appropriate cost-effectiveness thresholds for Zimbabwe.

### AMR surveillance

EGASP recommends the collection of at least 100 gonococcal isolates per year per country. Assuming a prevalence of NG of 3% among pregnant women, only 30 women with NG will be identified. To supplement the number of isolates obtained from ANC participants, men aged 18 years and above presenting with urethral discharge to the study clinics who provide written informed consent will be recruited. Consecutive sampling will be performed as resources allow, with the antenatal testing strategy taking priority.

Men with urethral discharge have been selected as the yield of a positive gonococcal culture will be high. Assuming an estimate of 50% NG prevalence among symptomatic men,[37] we anticipate recruiting approximately 140 men with urethral discharge. We therefore estimate isolation of 70 gonococcal isolates from men which, in addition to 30 isolates from women, will give 100 NG isolates in total. If the prevalence of NG is

lower than predicted, recruitment of men with urethral discharge will continue until 100 NG isolates have been obtained.

Data on age, sex, risk factors for STIs, including number of recent sexual partners and gender of sexual partners, current STI symptoms, recent antibiotic use, treatment provided at the PHC and previous STI diagnoses including HIV, will be collected using an interview-administered questionnaire. Nurse-collected urethral swabs will be collected from all enrolled men, and they will be managed syndromically according to national guidelines as part of routine care.

### Testing procedures for AMR

WHO EGASP guidelines and standard operating procedures will be adapted for use with pregnant women, and otherwise followed for AMR testing for NG.[38] Nurse-collected cervical samples will be obtained using ESwabs (COPAN Diagnostics) from women initially found to be positive for NG using the Xpert assay. Speculum examinations will be conducted by nurses with many years of experience performing these examinations, using plastic disposable speculums. Samples will be plated onto the InTray GC (Biomed Diagnostics) in-vitro device, as will urethral samples collected from men. These will be subsequently incubated in the laboratory and minimum inhibitory concentrations of ceftriaxone, cefixime, azithromycin and ciprofloxacin will be determined using Etest (bioMérieux) and interpreted using the European Committee on Antimicrobial Susceptibility Testing standards.[39]

**Table 3** Minimum detectable ORs for factors associated with presence of STIs

| Sample size* | STI prevalence | Precision | Smallest OR detected at 80% power | Smallest OR detected at 90% power |
|---|---|---|---|---|
| 896 | 20% | 2.6% | 1.74 | 1.89 |
| | 25% | 2.8% | 1.67 | 1.80 |
| | 30% | 3.0% | 1.63 | 1.75 |

*Assuming a prevalence of a risk factor of 20% among those without STIs.
STIs, sexually transmitted infections.

## Sample size calculations

There are limited data on STI prevalence in pregnant women in Zimbabwe. A prevalence of curable STIs between 32.0% and 37.0% has been reported in South Africa and Zambia.[12–16] Our recent studies among female youth in Harare have demonstrated combined CT/NG prevalence between 18.2% and 19.5%.[18 19] The prevalence of hepatitis B in pregnant women in South Africa has ranged between 3.1% and 5.3%.[20–22] Therefore, a conservative estimate of composite prevalence of curable STIs and hepatitis B is 30.0%. With a desired precision of 3% and alpha of 0.05, a sample size of 896 is required. To allow for invalid test results, 1000 pregnant women will be screened. The minimum detectable ORs for factors associated with the presence of STIs at different composite prevalence of STIs are shown in table 3.

## Statistical analysis

The primary outcome measure is the composite prevalence of CT, NG, TV, syphilis and hepatitis B in this population.

Other outcome measures include:
1. Individual prevalence of each STI (CT, NG, TV, syphilis, hepatitis B and HIV).
2. Uptake of testing.
3. Uptake of treatment.
4. Uptake of partner notification.
5. STI yield (number of participants with a positive STI result/total number of eligible individuals).
6. Prevalence of AMR to ceftriaxone, cefixime, azithromycin and ciprofloxacin in NG isolates.
7. Prevalence of preterm birth, miscarriage and low birth weight.

Categorical variables will be described using frequencies and percentages. Continuous variables will be described using either mean (SD) or median (IQR) for normally distributed and non-normally distributed data, respectively. Multivariable logistic regression will be used to assess factors associated with presence of STIs. Clustering will be adjusted for at clinic level. Logistic regression will also be used to assess the relationship between STI diagnosis and birth outcomes.

## Data management procedures

Quantitative data will be collected using electronic case report forms on tablet computers using Open Data Kit software, with range restrictions and dropdown menus to minimise data entry errors. CT/NG results recorded on tablet computers will be cross-checked with the readout from the GeneXpert device. Data will be managed and cleaned using STATA (StataCorp, Texas, USA). Interviews and focus group discussions will be audio recorded and transcribed verbatim. Anonymisation of transcripts will be performed once translation and transcription have been completed.

All data will be stored in password-controlled databases, and all data will be encrypted. All data collected will be anonymised using a unique study ID. Any identifiable data (eg, locator forms and consent forms) will be stored in secure, locked facilities with access limited to the study team.

## Data sharing

The informed consent procedure will clarify the sharing of anonymised data, either via a public data repository or by directly sharing with other researchers.

At the time of publication of research, the subset of data required for the purposes of verifying research findings will be made available for sharing and will be placed in Data Compass (the London School of Hygiene & Tropical Medicine institutional research data repository—accessible at https://datacompass.lshtm.ac.uk/). This repository will enable direct download of records with codebooks to enable replication of the data analyses. A more complete sharing of data with any research group requesting access to individual data records will be done 12 months after publication. At this point, all data and study tools will be made available through Data Compass. Data for sharing will be de-identified prior to release. Details of how to access data will be published with each study publication.

## Patient and public involvement

Initial formative work will include dialogue and input from service users to help refine the testing strategy and data collection tools. Findings will also be disseminated through study clinics.

## Ethics and dissemination

The study protocol has been approved by the MRC of Zimbabwe (MRCZ/A/2899), the Biomedical Research and Training Institute Institutional Review Board (AP176/2022) and the London School of Hygiene & Tropical Medicine Research Ethics Committee (26787). The completed Standard Protocol Items: Recommendations

for Interventional Trials checklist can be found in the online supplemental material.

Written informed consent to participate in the study will be obtained in the preferred language of the potential participant (English or Shona). Specific consent will also be sought for storage of samples and the sharing of anonymised data via a public data repository. Participants will not be identifiable from this information. An example participant informed consent form for enrolment into the main STI testing study can be found in the online supplemental material.

In Zimbabwe, individuals who are under 18 years of age and pregnant are considered emancipated minors. Therefore, independent informed consent will be obtained from pregnant minors.

For any participants who are under 16 years of age and pregnant, we will consider on an individual basis whether further input is needed regarding child protection. This may include discussion of the case with a multidisciplinary team and possible referral to social services. If these instances do arise, we aim to integrate into existing clinic processes as much as possible.

Results will be submitted to open-access peer-reviewed journals, presented at academic meetings and shared with participating communities and with national and international policymaking bodies.

## DISCUSSION

There is mounting evidence that syndromic management is not an effective method for the control of STIs.[14 40 41] However, evidence is required by national and international policymakers to inform how to transition towards diagnostic testing for STIs. We anticipate high prevalence of STIs among pregnant women in this study, thus providing additional evidence that new strategies are required for control of STIs in Zimbabwe. The data on acceptability, feasibility and cost-effectiveness will provide guidance on how best to implement new strategies for a package of integrated diagnostic testing in ANC. Furthermore, this study will provide baseline data for the design of future testing strategies, implemented at a larger scale and potentially as part of a cluster-randomised trial to demonstrate efficacy.

The IPSAZ Study has several strengths. The large sample size ensures that STI prevalence in the antenatal population in Harare will be estimated with high precision. The use of mixed methods for the process evaluation will enable quantitative and qualitative data to complement and inform each other. Furthermore, the economic evaluation will inform policymakers about whether introducing such a testing programme represents a cost-effective use of health resources in this and similar contexts. Finally, we will be following adapted EGASP standard procedures for gonococcal culture and AMR testing. EGASP guidelines do not currently include provision for sample collection from pregnant women and are focused on symptomatic men. Our experience will therefore inform the potential

expansion of EGASP guidelines to include pregnant women, and to consider the use of antenatal networks as a platform for gonococcal surveillance for future national EGASP programmes.

We acknowledge some limitations. There is no formal outcome evaluation comparing birth outcomes between those who received the intervention and those who did not. This is an important research question. Although treatment of STIs is likely to prevent complications such as pelvic inflammatory disease, there is conflicting evidence on whether provision of treatment for STIs during pregnancy prevents adverse birth outcomes.[42–44] Future studies need to investigate the clinical effectiveness of scaling up of STI testing and treatment in ANC.

The testing strategy will be delivered by members of a dedicated study team, which may not be representative of how testing would integrate into existing governmental clinical services, especially given that resources and staff are severely constrained in Zimbabwe. However, the mixed-methods approach to the process evaluation will provide supporting information to contextualise the findings.

Finally, the study will be conducted in urban PHCs in Harare province. As a result, the results will likely be generalisable to urban centres in Southern Africa, but less so to rural areas in this region.

In summary, the IPSAZ Study will provide important data to develop an integrated screening package for STIs in ANC in Southern Africa. The epidemiological data, process evaluation and economic evaluation will all help inform sustainability and scalability, in order to provide evidence-based policy recommendations. Additionally, the collection of gonococcal AMR data will also be used to inform national STI treatment guidelines in Zimbabwe.

**Author affiliations**
[1]Department of Clinical Research, London School of Hygiene & Tropical Medicine, London, UK
[2]The Health Research Unit Zimbabwe, Biomedical Research and Training Institute, Harare, Zimbabwe
[3]Department of Global Health and Infection, Brighton and Sussex Medical School, Brighton, UK
[4]Department of Infectious Disease Epidemiology, London School of Hygiene & Tropical Medicine, London, UK
[5]Department of Global Health and Development, London School of Hygiene & Tropical Medicine, London, UK
[6]Skin & Genito-Urinary Medicine Clinic, Harare, Zimbabwe
[7]Internal Medicine Unit, Faculty of Medicine and Health Sciences, University of Zimbabwe, Harare, Zimbabwe
[8]AIDS and TB Unit, Ministry of Health and Child Care, Harare, Zimbabwe
[9]Hospital for Tropical Diseases, University College London Hospital, London, UK
[10]Division of Infection and Immunity, University College London, London, UK
[11]Division of Infectious and Tropical Medicine, Medical Centre of the University of Munich, Munich, Germany

**Contributors** KM conceptualised the IPSAZ Study, supervised by RF, KK and MM. KM developed the first draft of the paper, developed the study manual of operations and will be responsible for study implementation. KM and TB are responsible for data management. CDC, ED, CRSM-Y, DB, JT, VS, TB, FN, LK, OM and AM provided guidance on study design and methodology. All authors reviewed the final draft of the manuscript.

**Funding** This work was supported by a Wellcome Trust grant awarded to KM (grant number 225468/Z/22/Z).

**Disclaimer** The funder of the study has had no role in study design, and will have no role in data collection, data analysis, data interpretation, writing of the report or decision to submit.

**Competing interests** None declared.

**Patient and public involvement** Patients and/or the public were involved in the design, or conduct, or reporting, or dissemination plans of this research. Refer to the Methods section for further details.

**Patient consent for publication** Not required.

**Provenance and peer review** Not commissioned; externally peer reviewed.

**ORCID iDs**
Kevin Martin http://orcid.org/0000-0001-6561-1353
Chido Dziva Chikwari http://orcid.org/0000-0003-1617-3603
Constance R S Mackworth-Young http://orcid.org/0000-0002-9725-7931
Victoria Simms http://orcid.org/0000-0002-4897-458X
Michael Marks http://orcid.org/0000-0002-7585-4743

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
