## [Reviewer comments · BMJ Open]

ARTICLE DETAILS

TITLE (PROVISIONAL)	Investigating point-of-care diagnostics for sexually transmitted infections and antimicrobial resistance in antenatal care in Zimbabwe (IPSAZ): protocol for a mixed methods study
AUTHORS	Martin, Kevin; Dziva Chikwari, Chido; Dauya, E; Mackworth-Young, Constance; Bath, David; Tucker, Joseph; Simms, Victoria; Bandason, T; Ndowa, Francis; Katsidzira, Leolin; Mugurungi, Owen; Machiha, Anna; Marks, Michael; Kranzer, Katharina; Ferrand, Rashida

VERSION 1 – REVIEW

REVIEWER	Mohamed, Yasmin Burnet Institute
REVIEW RETURNED	09-Feb-2023

GENERAL COMMENTS	This is a well-designed and important study; thank you for the opportunity to review this protocol. The authors have clearly described their rationale and methodology, including outlining the strengths and limitations of the study. The methods are comprehensive (particularly the use of mixed methods combined with an economic evaluation) and appropriate to the research question. I have a few suggestions to improve the clarity of the manuscript: 1. It would be helpful to make it clearer in the abstract and the introduction that testing for HIV and syphilis are already part of routine antenatal care, so that it is more obvious which tests are specifically part of the evaluation2. I would suggest including which POC test will be used for each STI (e.g., Xpert tests for CT/NG) in the introduction or earlier in the methods3. Are there any more recent estimates of the global STI burden since 2016?4. I think it would be worth highlighting that STIs are more commonly asymptomatic in women, particularly given the focus of the study.5. The first time the authors use the term “AMR” (third paragraph of the introduction) it should be written out in full.6. In the economic evaluation, how will the clinical endpoint of infertility be assessed in pregnant women?7. I understand that syphilis prevalence is part of the primary outcome measure. Will HIV prevalence also be reported? Given that syphilis POC testing is already part of routine care, will it also be part of the process evaluation?
--

REVIEWER	Vallely, Lisa
-----------------	---------------

	UNSW, Kirby Institute -Faculty of Medicine
REVIEW RETURNED	14-Feb-2023

GENERAL COMMENTS	Many thanks for the opportunity to review this protocol for this important area of work to add to the body of evidence of POC testing in pregnancy. I have only a couple of comments for consideration.  1. In table 1 (P9) you describe cervical swabs but on P10 you say there will be 3 self collected OR provider collected VAGINAL swabs. Please could you clarify. Assume self collected will be vaginal swabs. 2. On P 16, you state that you will collect cervical swabs on NG positive women for AMR using Cobas. Will there be additional consent for this or will it be included in the initial consent when women enrol? 3. Assuming that for cervical swabs women will require a speculum examination. Could you state that this will be done by obs/ gynae specialist given the risk associated with speculum examination in pregnant women. 4. In reporting preterm birth, how will this be assessed. We know ultrasound in first trimester is the best predictor for assessing GA and comparing against the LMP. How will prematurity be identified, assuming most women won't have an ultrasound. Will preterm birth be based on LMP alone, as this is likely to be unreliable. Maybe you could add a line in methods to say that PTB will be based on LMP or USS, just for clarification. 5. Am wondering why you are not testing for BV, given it is such a quick and easy test. I know, strictly speaking it is not an STI, but the data would add to the body of evidence we need for this genital infection among pregnant women.
---

VERSION 1 – AUTHOR RESPONSE

Reviewer 1

- This is a well-designed and important study; thank you for the opportunity to review this protocol. The authors have clearly described their rationale and methodology, including outlining the strengths and limitations of the study. The methods are comprehensive (particularly the use of mixed methods combined with an economic evaluation) and appropriate to the research question.

Thank you for your comments.

1. It would be helpful to make it clearer in the abstract and the introduction that testing for HIV and syphilis are already part of routine antenatal care, so that it is more obvious which tests are specifically part of the evaluation

To make this more clear, we have amended the abstract to read:

"Alongside routine HIV and syphilis testing, participants will be offered an integrated screening package including testing for Chlamydia trachomatis (CT), Neisseria gonorrhoeae (NG), Trichomonas vaginalis (TV), and Hepatitis B."

We have also reworded the following sentence in the introduction for clarity:

“As point-of-care (POC) testing for HIV and syphilis has already been integrated into ANC services, this provides a platform for further STI testing and management and potentially enhances operational feasibility.”

2. I would suggest including which POC test will be used for each STI (e.g., Xpert tests for CT/NG) in the introduction or earlier in the methods

We believe that the methods is the most appropriate section of this paper for this information, as it deals directly with the procedures to be used in the study. The introduction aims to provide a background to the study, without getting into specific details of which tests we will use. The tests to be used are provided at the beginning of the “study procedures” section of the methods. We don't think it would be appropriate to move this information earlier as it is not relevant to “study design and setting” or “study population and recruitment”.

3. Are there any more recent estimates of the global STI burden since 2016?

2020 estimates from within the WHO global health sector strategies for HIV, viral hepatitis, and STIs, for 2022-2030, have now been provided.

4. I think it would be worth highlighting that STIs are more commonly asymptomatic in women, particularly given the focus of the study.

The following sentence has been amended to reflect this:

“This is problematic as the majority of curable STIs are asymptomatic, particularly in women, and are missed by syndromic management.”

5. The first time the authors use the term “AMR” (third paragraph of the introduction) it should be written out in full.

First use of the term “AMR” is in the second paragraph, with the full term written out.

6. In the economic evaluation, how will the clinical endpoint of infertility be assessed in pregnant women?

Due to the relatively short timeframe of the study, the incidence of clinical endpoints will be modelled based on existing literature. For adverse birth outcomes, this modelling will be supported by primary data collected during the study. We have added the following sentence to make this more clear:

“Incidence of clinical endpoints will be estimated from existing literature, with such estimates for adverse birth outcomes being supported by primary data collection.”

7. I understand that syphilis prevalence is part of the primary outcome measure. Will HIV prevalence also be reported? Given that syphilis POC testing is already part of routine care, will it also be part of the process evaluation?

HIV prevalence will be reported. We have amended the list of outcome measures to make this more clear.

As HIV and syphilis testing are part of the testing intervention, they will both be considered by the process evaluation, and particularly how well they integrate with the additional tests.

Reviewer: 2

Dr. Lisa Vallely, UNSW

Comments to the Author:

Many thanks for the opportunity to review this protocol for this important area of work to add to the body of evidence of POC testing in pregnancy. I have only a couple of comments for consideration.

Thank you for your comments

1. In table 1 (P9) you describe cervical swabs but on P10 you say there will be 3 self collected OR provider collected VAGINAL swabs. Please could you clarify. Assume self collected will be vaginal swabs.

3 vaginal swabs will be collected, which can be either self- or provider-collected. A single provider-collected cervical swab will only be collected from participants who test positive for gonorrhoea. For clarification, the schedule of events (table 1) has been amended to include collection of the 3 vaginal swabs.

2. On P 16, you state that you will collect cervical swabs on NG positive women for AMR using Cobas. Will there be additional consent for this or will it be included in the initial consent when women enrol?

To clarify, we are collecting cervical samples using ESswabs® (COPAN Diagnostics Inc.), rather than Cobas.

Consent for collection of cervical swabs from NG positive women for AMR is included in the initial consent process. Participants can decide to withdraw at any time, or refuse specific procedures. The main consent form is now included as supplemental material.

3. Assuming that for cervical swabs women will require a speculum examination. Could you state that this will be done by obs/ gynae specialist given the risk associated with speculum examination in pregnant women.

Speculum examinations will be conducted by nurses with many years of experience performing these examinations, using plastic disposable speculums. The manuscript has been amended to include this information

4. In reporting preterm birth, how will this be assessed. We know ultrasound in first trimester is the best predictor for assessing GA and comparing against the LMP. How will prematurity be identified, assuming most women won't have an ultrasound. Will preterm birth be based on LMP alone, as this is likely to be unreliable. Maybe you could add a line in methods to say that PTB will be based on LMP or USS, just for clarification.

Pre-term birth will be based on LMP alone. The following sentence has been added to the methods to reflect this:

“Estimated due date, which will be compared with actual birth date to determine prematurity, will be based on last menstrual period.”

5. Am wondering why you are not testing for BV, given it is such a quick and easy test. I know, strictly speaking it is not an STI, but the data would add to the body of evidence we need for this genital infection among pregnant women.

Thank you very much for your comment. It is something that we spent some time considering. Our thought process was that, as the evidence is mixed regarding treatment of asymptomatic BV, we would only have considered screening participants with symptoms. With that strategy, we expected yield to be low, and decided against its inclusion. We also wanted to keep the intervention relatively streamlined, with all participants receiving the same initial batch of tests.

We agree that more data is needed for BV, and will definitely consider its inclusion for future studies.